# MViT: A vision transformer with fractal path reordering and dynamic positional encoding

**Bomin Liu, Linjun He, Yan Zhu** *

School of Design and Art, Shanghai Dianji University, Shanghai, China

* Zhuy@sdju.edu.cn

## Abstract

Vision Transformers have demonstrated remarkable performance in image classification and structural modeling; however, fixed patch partitioning and static positional encoding often disrupt spatial continuity, thereby limiting their ability to represent rotated structures and irregular boundary regions. To address these limitations, we propose the Moore-curve Vision Transformer (MViT), a Vision Transformer (ViT) framework based on a recursive Moore curve. The proposed framework comprises three key components. First, a multi-order fractal mapping is employed to optimize patch reordering and enhance the spatial coherence of the token sequence. Second, a 7×7 dynamic partitioning template together with a boundary compensation algorithm jointly optimizes dense structural representation and resolution adaptability. Third, a period-aware positional encoding module integrates fractal periodic parameters with convolutional features to align positional embeddings with the fractal traversal pattern. This design significantly enhances the structural adaptability of the model to complex image layouts. Experimental results show that MViT improves classification accuracy over ViT-B/16 by 0.52% and 0.31% on the CIFAR-100 and ImageNet-21k datasets, respectively, while also achieving noticeable improvements in PSNR and SSIM. Ablation and rotational perturbation experiments further confirm its robustness to rotation and localized focus variations. Moreover, MViT exhibits strong structural compatibility, maintaining stable performance across different Transformer backbones and diverse visual tasks.

## 1 Introduction

In recent years, Vision Transformers [1] have significantly improved the performance of tasks such as image classification and object detection by introducing the Transformer architecture from natural language processing into the field of computer vision [2,3]. However, current ViT models still exhibit two major limitations. First, the use of fixed-size patching and linear sequence modeling for global dependency tends to disrupt local spatial continuity [4], leading to semantic fragmentation [5]. Second,

**Data availability statement:** The data underlying the results presented in the study are available from: https://www.modelscope.cn/datasets/timm/imagenet-22k-wds; https://figshare.com/authors/Linjun_He/22524461; https://github.com/visipedia/inat_comp/tree/master/2021; https://github.com/CSAILVision/places365.

**Funding:** This work was funded by the Industry-university Cooperation and Collaborative Education Project of the Ministry of Education under Grant 231003221260825, and in part by the 2024 Graduate Education Reform Project of Shanghai Dianji University under Grant A102252401102029.

**Competing interests:** The authors have declared that no competing interests exist.

the structure of absolute positional encoding is difficult to adapt to spatial transformations such as rotation and symmetry, and it easily causes information truncation on images with non–power-of-two resolutions [6].

These problems reduce adaptability to local structures and spatial variations. Recent improvements have mainly focused on three directions: introducing spatial inductive bias to enhance local modeling capability, constructing structure-aware window mechanisms, and exploiting fractal curves to strengthen the spatial continuity of positional encoding. For example, the Data-efficient Image Transformer (DeiT) [7] improves data efficiency through knowledge distillation, and the Shifted Window Transformer (Swin Transformer) enhances spatial awareness through local window attention. The Geometric-Fractal-based Positional Encoding Vision Transformer (GFPE-ViT) [8] employs a positional encoding based on the Hilbert curve to alleviate the discontinuity of linear patch ordering, while the Hilbert Curve-based Point-Net (Hilbert-PointNet) [9] successfully applies the Hilbert curve to point cloud modeling, demonstrating the effectiveness of such curves in representing complex spatial structures. These methods have enhanced the performance of visual Transformers in maintaining spatial structures and modeling geometric relationships, demonstrating consistent structural expression across different tasks. For example, in medical imaging research, the U-shaped convolutional neural network (U-Net) model has been employed for lung signal slicing, enabling precise modeling of Coronavirus Disease 2019 (COVID-19) lesions [10]. In smart agriculture, visual Transformers have been used for the automated detection of pig group interaction behaviors., indicating that such architectures maintain stable feature representations and robust generalization in complex visual environments [11].

Although existing studies have made progress in local structure modeling and positional encoding, three major limitations remain. First, fixed sequential traversal paths weaken spatial adjacency between patches and reduce local continuity. Second, most methods rely on input resolutions that are powers of two, which makes it difficult to maintain consistent boundary alignment for irregular image sizes. Third, conventional absolute or relative positional encodings lack periodic modulation, leading to phase mismatch and index mapping mismatch under rotations, symmetries, or scale changes, thereby limiting structural representation and geometric generalization in complex visual tasks. To address these issues, this study proposes MViT, a recursive framework based on fractal mapping that jointly optimizes dynamic curve mapping and positional encoding to enhance local structure modeling and spatial awareness. The main contributions of this work are as follows:

1. A multi-order path construction mechanism is proposed, which employs recursive Moore curves to optimize the traversal order of image patches. This design provides a foundation of spatial continuity that enhances structural consistency modeling.
2. A period-aware positional encoding module is developed based on the inherent cyclic characteristics of the path structure. By integrating fractal period parameters with shallow convolutional features, this module enables adaptive positional embedding under structural alignment.

3. An adaptive $7 \times 7$ partitioning template and a boundary compensation strategy are introduced to improve the adaptability of the path mechanism to nonstandard image sizes, thereby strengthening structural preservation and encoding continuity at image boundaries.

These three components collectively constitute the core architecture of the Vision Transformer. The Moore path construction and the period-aware positional encoding modules exhibit high adaptability and can be seamlessly integrated into different models based on Transformers without modifying the attention mechanism. On the CIFAR-100 and ImageNet-21k datasets, MViT achieves classification accuracies of 93.82% and 84.08%, respectively, surpassing the baseline ViT-B/16 by 0.52% and 0.31%. Ablation experiments demonstrate that periodic modulation in positional encoding improves the focus of the attention mechanism on structural regions, while rotation perturbation experiments show a 12.6% reduction in recognition error for rotated targets. Moreover, MViT achieves a 30% faster convergence rate than ViT-B/16 on datasets of small scale, verifying its computational efficiency.

The remainder of this paper is organized as follows. Sect 2 reviews recent advances in Vision Transformers, analyzing patching strategies, positional encoding methods, and the foundations and challenges of fractal modeling. Sect 3 presents the MViT framework, detailing the Moore-curve path construction, the period-aware positional encoding mechanism, and the boundary compensation strategy. Sect 4 describes the experimental setup, including performance evaluation across multiple datasets, structural modeling assessment, ablation studies, training efficiency, parameter sensitivity, compatibility across different backbones, lightweight evaluation, visualization analysis, and method discussion. Finally, Sect 5 concludes the paper by summarizing the design advantages and experimental outcomes and discussing potential future research directions.

## 2 Related work

This section reviews recent advances in Vision Transformers, including image sequence construction, positional encoding, and fractal path modeling. It further examines the structural adaptability, geometric modeling capability, and application limitations of existing approaches, thereby providing a theoretical foundation for the model design presented in the following sections.

### 2.1 Image sequence construction and positional encoding mechanism optimization

Vision Transformers achieve cross-regional spatial modeling by dividing images into patches and constructing token sequences. However, conventional sequence generation methods typically adopt fixed-size patching and static positional encoding strategies, which fail to preserve spatial continuity and cannot adapt to geometric transformations such as rotation and symmetry. As a result, these limitations reduce the ability of the model to represent complex visual scenes. Recent studies have therefore focused on improving both the patch partitioning structure and the positional information encoding mechanism to enhance the consistency and flexibility of structural modeling.

The patching strategy constitutes a fundamental component for achieving image serialization in Vision Transformers. The manner of patch division directly affects the preservation of local structures and the understanding of spatial context. A well-designed partitioning mechanism can improve structural modeling while maintaining computational efficiency. The original ViT [1] converts an image into a token sequence through fixed-size patches with a size of $16 \times 16$. However, fixed partitioning often disrupts local continuity and leads to semantic fragmentation. To optimize patching strategies, various improved methods have been proposed. Swin Transformer [2] introduces a hierarchical window attention mechanism that reduces computational complexity through localized modeling, although fixed windows limit multi-scale adaptability. Deng et al. [12] proposed the TransBridge architecture, which integrates context-aware patching to improve detection performance in complex scenarios. Axial attention [13] performs independent modeling along row and column axes, reducing computational complexity to linear complexity but compromising long-range dependency modeling. A patch reordering

method based on an improved Peano curve [14] enhances local adjacency by optimizing patch traversal through spatial filling, but exhibits limited adaptability to nonuniform textures. Han et al. [15] developed a graph-structured Transformer that introduces a spatially adaptive mechanism to strengthen local modeling, albeit with increased computational cost. The design of patching strategies must balance spatial continuity, computational efficiency, and scale adaptability in order to achieve effective structural modeling.

Positional encoding is an essential mechanism that supplements spatial ordering information and compensates for the lack of explicit sequential modeling in Transformer architectures. In visual tasks, dynamic positional encoding improves the modeling of spatial dependency and geometric transformation by combining image content with spatial structural features. However, conventional absolute positional encoding has limited adaptability to spatial transformations and semantic diversity in complex visual scenes.To mitigate this limitation, the Detection Transformer (DETR) [3] introduces learnable positional embeddings to improve detection performance, although the positional representation it produces is still static. DeiT [6] employs knowledge distillation to strengthen positional information modeling, but it does not explicitly integrate image semantics, which restricts the consistency of the learned representation. Liu et al. [16] present a dynamic positional encoding mechanism that uses local semantic cues to enhance structural awareness and boundary modeling. Hong et al. [17] propose a formulation based on relative positional displacement, and Swin Transformer applies a positional bias defined within local windows so that the model remains robust under image rotation and translation. Wu et al. [18] further develop an adaptive bias strategy that improves the modeling of local semantics and fine spatial structure.Although these methods have achieved clear progress, they still suffer from high computational cost and limited generalization across tasks. The approach proposed in this work combines a Moore traversal path with a periodic positional encoding module(PPE). The proposed module encodes tokens along the Moore path through a periodic parameter T and controlled phase modulation, which enforces coherent positional progression. This design acts as a lightweight component at the input stage and can be inserted into different ViT backbones without any change to the attention layers or the task heads. In this way, it improves robustness to local structural patterns and rotated content.

## 2.2 Application of fractal geometry in computer vision

Fractal geometry provides a theoretical foundation for local structural modeling by describing recursive geometric patterns and the ability of certain curves to fill a region continuously without gaps. Classical examples include the Hilbert curve and the Peano curve, which have been widely used in visual computing. Yang et al. [8] proposed a method that uses the Hilbert curve to enhance local continuity and spatial reconstruction in image structures. Ma et al. [19] reported that the Hilbert–Huang transform has high computational cost in signal decomposition at higher order, which restricts its practical use in structural modeling. The Peano curve supports flexible recursive subdivision, and it has been used to build a patch reordering strategy [14]; however, such a strategy shows limited adaptability when the texture is not uniform [20]. Chen et al. [21] combined structural and textural information across multiple scales and improved local detail modeling in image fusion. The Moore curve [9], which can be viewed as an extension of the Hilbert curve, achieves continuous coverage of the plane through mirror symmetry and offers a new perspective for patch organization in visual Transformer architectures. Zheng et al. [22] integrated fractal descriptors into medical image analysis and reported improved lesion detection. Existing methods that rely on fractal structure still face challenges such as difficult parameter tuning and high computational cost, which indicates the need for a lightweight and efficient modeling framework.

## 2.3 Non-power-of-two patch partitioning

In practical applications, image dimensions often do not follow power-of-two patterns, making direct adaptation of conventional Vision Transformers difficult. Classical fractal curves such as the Hilbert curve and the Moore curve depend on block sizes defined by powers of two, which limits their ability to process irregular image resolutions and often causes

boundary information loss [23]. GFPE-ViT employs a 7 × 7 partitioning template to accommodate non-power-of-two resolutions, yet the fixed template cannot dynamically adjust the fractal order, thereby reducing flexibility [24]. To address this issue, Yeom et al. [25] introduced a joint structural and positional encoding mechanism that improves adaptability to irregular spatial layouts. Wang et al. [26] proposed a hybrid fractal strategy that integrates low-order filling and boundary interpolation to enhance adaptability. However, as dynamic prediction relies on hardware acceleration [27] and hybrid strategies tend to introduce noise artifacts [28], a tradeoff among flexibility, efficiency, and reconstruction quality remains necessary.

## 3 Method

This section presents the MViT architecture, which integrates fractal geometry with mechanisms that are sensitive to structural information to form a unified design framework covering patch arrangement, positional encoding, and boundary modeling. The core idea is to construct a structurally continuous sequence of patches by means of a recursive Moore curve. This sequence guides the partitioning order and in turn supports both periodic positional embedding and adaptive boundary handling.

### 3.1 MViT overall architecture

Fig 1 illustrates the overall architecture, which comprises three core modules. (1) Moore curve-based Image patch reordering mechanism,designed to maintain spatial topological continuity; (2) Period-aware positional encoding module, providing structurally consistent embedded representations; and (3) Structurally adjustable boundary compensation mechanism, adapting to arbitrary-sized image inputs.

Input images undergo shallow feature extraction via a convolutional network and are subsequently divided into $P \times P$ patches. Subsequently, a path generator reorders the patch sequence based on multi-order Moore curves, constructing structurally continuous sequences. The period-aware positional encoding module combines path period parameter $T$ with feature vector $F \in R^{N \times C}$, generating embedding vector $E$.

$$E = \sigma(W_1 F + W_2 e_T) \tag{1}$$

where $W_1$ and $W_2$ are linear projection matrices, $\sigma(\cdot)$ is an activation function, and $e_T$ is a periodic encoding vector. This module ensures that the embedded representation aligns with the path structure. Considering non-standard resolution image inputs, a boundary compensation mechanism is introduced. It employs a 7 × 7 Moore template for structured padding of residual regions, thereby enhancing positional information completeness. All embedded patches are then fed into a standard Transformer encoder for feature modeling and classification tasks.

### 3.2 Recursive moore curve partitioning strategy

In conventional ViT architectures, images are partitioned into fixed-size patches and linearly flattened in a row-major order. However, such linear unfolding disregards the spatial connectivity and positional structure within the original image, potentially leading to local structural discontinuities and diminished modeling capability, especially when handling symmetric structures, rotational textures, and boundary regions. To address these issues, MViT designs a Moore curve-based patch reordering mechanism, which generates spatially continuous and structure-aware patch arrangement paths through three strategies: multi-order path construction, direction enhancement, and repetitive traversal.

The recursive construction process of Moore curves from first to third order is illustrated in Fig 2. The curve utilizes four quadrants as its fundamental unit, recursively copying and rotating the original path at each iteration to form a traversal path characterized by spatial connectivity and structural consistency. This inherent spatial connectivity and structural consistency ensure that adjacent patches in the image maintain spatial proximity when unfolded into a sequence,

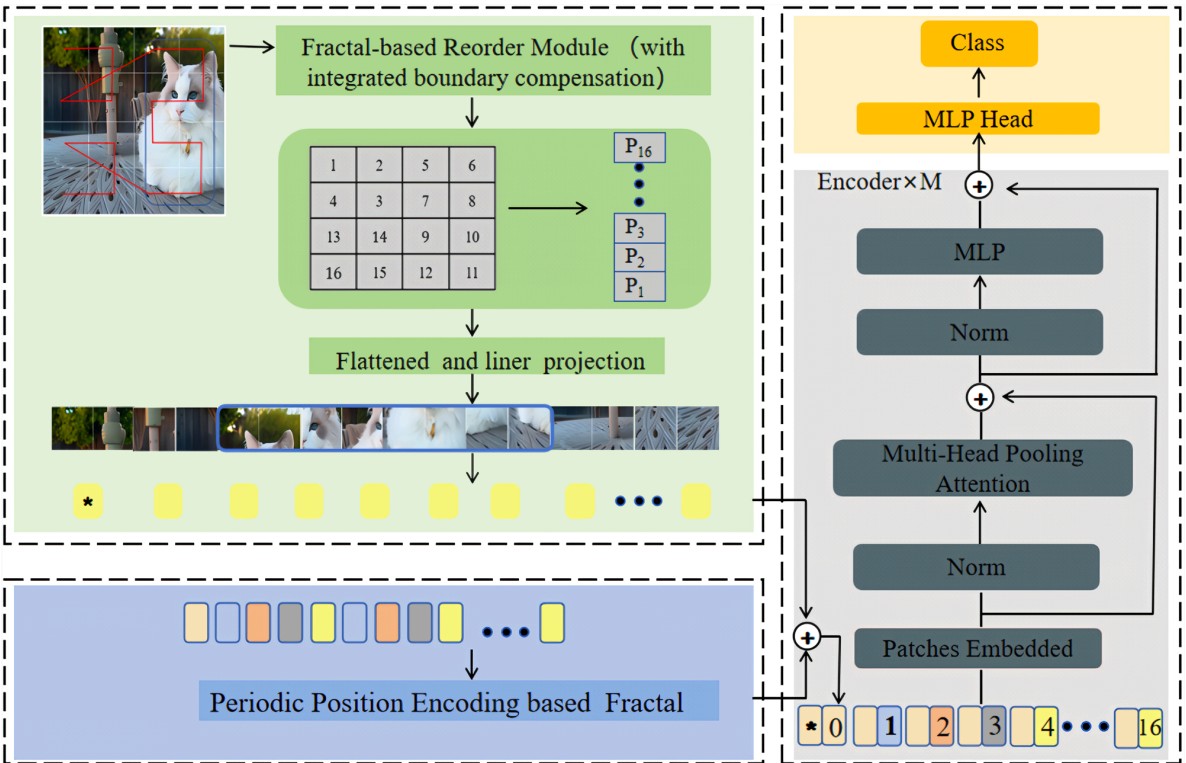

**Fig 1. Overall architecture of MViT.** Input images are processed by the fractal reordering module (integrating boundary compensation) to generate spatially continuous patch sequences. These sequences are then flattened and linearly projected to obtain initial token vectors. These vectors are summed with period-aware positional encodings and a classification token, subsequently fed into a multi-layer Transformer encoder for structural feature extraction, and finally processed by an MLP classification head to complete class prediction.

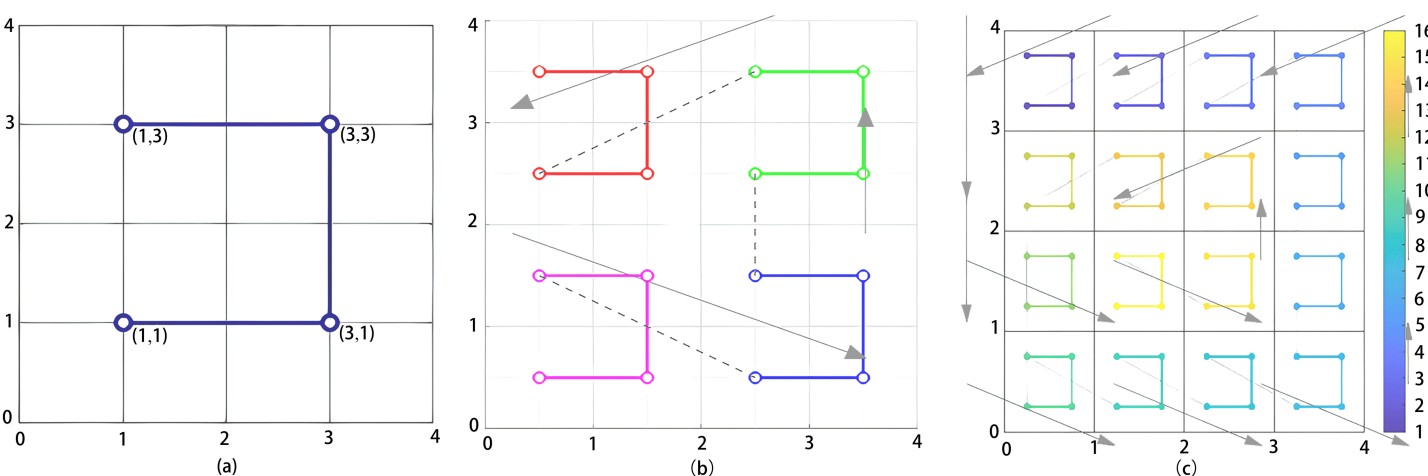

**Fig 2. Recursive construction of Moore curves at different orders.** The construction of Moore paths across multiple levels effectively preserves spatial continuity. Fig 2a illustrates the Moore curve at the first level, Fig 2b corresponds to the second level, and Fig 2c shows the structure at the third level, revealing how recursive expansion progressively enforces structural consistency in patch sequencing.

thereby enhancing the Transformer input sequence's coherent representation capability for local structures and providing an alignable path structure foundation for subsequent period-aware positional encoding.

Standard Moore curve construction typically fixes the starting point and limits traversal to a single direction, which can lead to coverage bias for image edges or structures oriented in specific directions. To enhance the structural generalization capability of the path, MViT designs three path enhancement strategies, as illustrated in Fig 3. Fig 3(a) adjusts the starting point to the bottom right corner, increasing the perceptual density of the path in the lower right quadrant; Fig 3(b) rotates the entire path by 45 degrees, improving adaptability for modeling diagonal textures and structures; and Fig 3(c) applies a horizontal mirroring operation, enhancing the expressive power of the model for symmetrical targets. These three strategies respectively compensate for limitations such as bias in the path starting point, singular direction, and lack of symmetry. Their common goal is to expand the response coverage of the path for spatial structures, thereby improving the capability of the model in structural consistency modeling under multi-modal inputs. These operations maintain the connectivity and recursive form of the path while enhancing directional elasticity and perceptual robustness.

Additionally, image edge regions often have lower coverage during path traversal, leading to insufficient encoding of boundary information. To address this issue, Fig 4 illustrates the design of a repetitive traversal mechanism, which adds supplementary traversal sub-paths to edge regions during the path construction process. This mechanism enhances the weight of edge patches in the sequence through redundant paths, thereby strengthening the modulation capability of period-aware positional encoding at boundaries and effectively alleviating the weakening of positional information.

Mapping the recursive path structure to linear sequence positions requires the definition of a unified recursive index function. Four types of quadrant recursive patterns in Moore path construction are shown in Fig 5, each corresponding to quadrant encoding $Q(x, y) \in \{0, 1, 2, 3\}$. The index mapping formula is defined as follows.

$$I(x,y,n) = 4 \cdot I\left(\left\lfloor \frac{x}{2} \right\rfloor, \left\lfloor \frac{y}{2} \right\rfloor, n - 1\right) + Q(x, y) \tag{2}$$

where $I(x, y, n)$ represents the one-dimensional sequence index of coordinates $(x, y)$ in the path of order $n$; $Q(x, y)$ is determined by its quadrant encoding, through recursively judging the belonging quadrant layer by layer and accumulating the positional index; and $\lfloor \cdot \rfloor$ represents the floor operation, used for coordinate reduction. The recursive structure of Moore

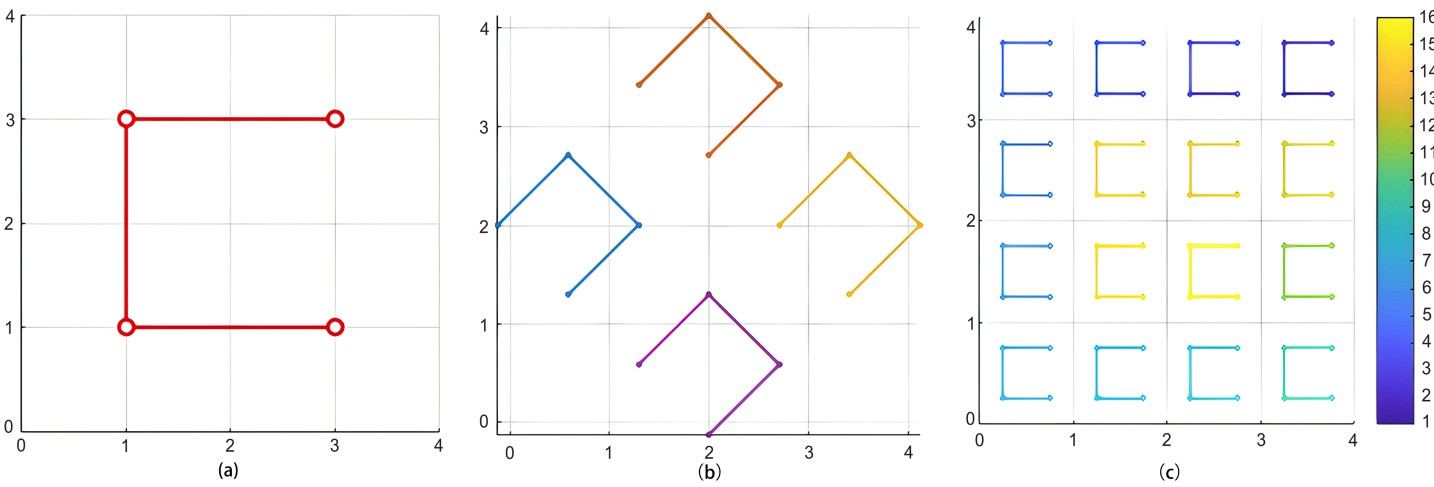

**Fig 3. Direction enhanced strategies for Moore curve paths.** Diagrams of direction enhancement for improved structural perception robustness are illustrated. Fig 3a presents the Moore curve at the first level, starting from the bottom right, Fig 3b depicts the curve at the second level with a rotation of 45 degrees, and Fig 3c shows the curve at the third level with horizontal mirroring.

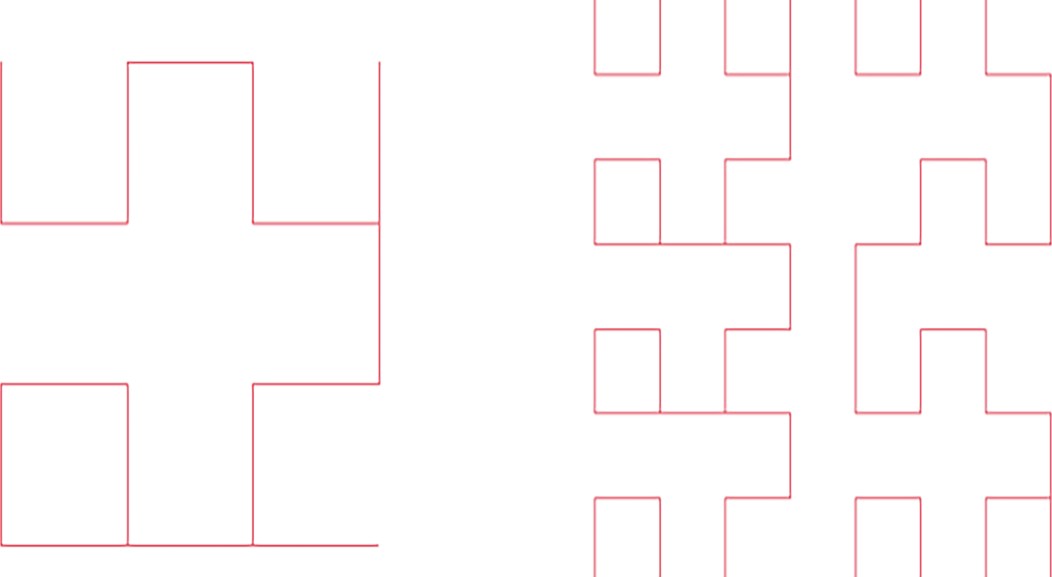

**Fig 4. Repeated traversal mechanism for boundary regions.** The left panel shows the repeated traversal mechanism at level two of the Moore path, where redundant subpaths are introduced along edges to increase local coverage and mitigate sparse patch responses. The right panel presents the level three structure with deeper recursion, enhancing boundary participation in periodic modulation and preserving embedding continuity.

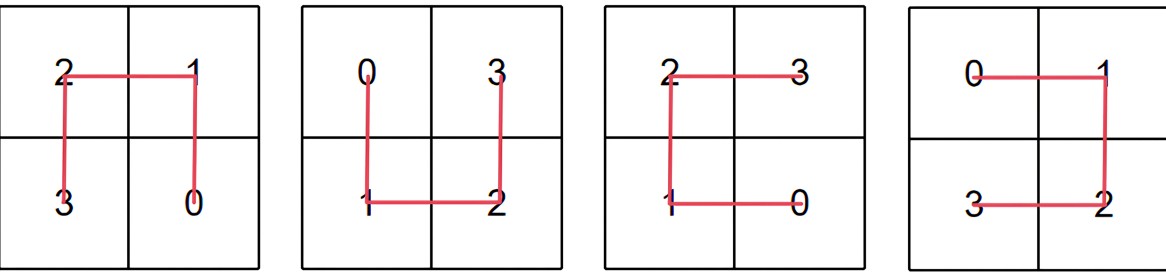

**Fig 5. Four recursive quadrant patterns used for path generation in Moore curve construction.**

curves, when unfolded into a linear sequence, requires the construction of a unified path indexing mechanism to maintain sequential consistency. Based on the aforementioned four-quadrant recursive relationship, a recursive path index generation algorithm (Algorithm 1) is designed, which can automatically generate traversal order for any given order. The algorithm possesses strong recursivity and structural consistency, and can flexibly adapt to path construction requirements of varying depths, providing a controllable sequence basis for subsequent positional encoding and patch reordering.

Fractal order $t$ is introduced to control the refinement level of the path structure. It determines the recursion levels of the Moore curve and its spatial coverage granularity. Each order construction divides the image into a grid, with the path length increasing exponentially with each division. A larger order enhances structure preservation but simultaneously increases the complexity of boundary compensation and periodic modulation. The default setting balances path representation and computational overhead, and its influence will be analyzed in the experimental section through sensitivity analysis.

**Algorithm 1 Converting patch 2D coordinates to sequence 1D position.**

```
Require:
   map_Moore = {
     a: {(0,0): (0, d), (0,1): (1, a), (1,0): (3, b), (1,1): (2, a)},
     b: {(0,0): (2, b), (0,1): (1, b), (1,0): (3, a), (1,1): (0, c)},
     c: {(0,0): (2, c), (0,1): (3, d), (1,0): (1, c), (1,1): (0, b)},
     d: {(0,0): (0, a), (0,1): (3, c), (1,0): (1, d), (1,1): (2, d)} }
   a, b, c, d denote four recursive quadrant patterns
   (X, Y) are 2D coordinates of the patch
   order is the recursion depth of the Moore curve
Ensure: pos – the 1D index in the linearized sequence
 1: current_pattern ← a
 2: pos ← 0
 3: for i from (order-1) downto 0 do
 4:    pos ← pos ≪ 2
 5:    if X&(1≪i) then
 6:       quad_x ← 1
 7:    else
 8:       quad_x ← 0
 9:    end if
10:    if Y&(1≪i) then
11:       quad_y ← 1
12:    else
13:       quad_y ← 0
14:    end if
15:    (quad_pos, next_pattern) ← map_Moore[current_pattern][(quad_x, quad_y)]
16:    pos ← pos | quad_pos
17:    current_pattern ← next_pattern
18: end for
19: return pos
```

### 3.3 Period-aware positional encoding module

During the recursive construction of the Moore path, a sequentially continuous patch arrangement is formed, and local regions present identifiable repeating patterns. This characteristic provides structural guidance for position modeling. Based on this property, a periodic positional encoding module is developed. A periodic parameter is introduced to regulate the distribution of the embedding sequence, which enhances the consistency of the model in representing symmetric regions and boundary structures. In the fractal construction of the Moore path, the path of order $t$ is divided into an $N \times N$ grid, and its traversal length is denoted as $L_t$. The periodic parameter $T$ corresponds to the order $t$. It describes the periodic pattern within the recursive path structure and adjusts the distribution of cosine and sine frequencies during the encoding stage. To ensure consistency between periodic modulation and path traversal patterns, the structural periodic parameter and path length satisfy an integer multiple relationship,

$$L_t = kT, \quad k \in \mathbb{N}^+ \tag{3}$$

where $t$ represents the recursion order of the Moore path, $L_t$ is the total number of patches covered by the path of order $t$, $T$ is the structural periodic parameter, and $k$ is a positive integer constant. This constraint maintains the correspondence between periodic modulation and fractal order, aligning the frequency distribution in positional encoding with the structural properties of the path. A periodic modulation embedding structure is then constructed, where pos denotes the index of a

patch in the Moore path and the positional embedding $e_{(}$pos$)$ is given by,

$$e(\text{pos}) = \left[\sin\left(\frac{\omega_k(\text{pos} + \phi)}{T}\right), \cos\left(\frac{\omega_k(\text{pos} + \phi)}{T}\right)\right]_{k=0}^{d/2-1} \tag{4}$$

where $d$ is the embedding dimension, $\phi$ is the learnable phase shift parameter, pos is the path index, and $T$ is the structural periodic parameter. The value of $T$ controls the frequency scale, while the phase parameter $\phi$ adjusts the alignment of the embedding position, generating a periodic representation consistent with the fractal path. In order to handle variations in image content and shallow features, a gating mechanism is applied to dynamically combine the periodic encoding with visual guidance features,

$$\begin{aligned} G &= \sigma\left(W_g \cdot F\right) \\ P &= G \odot E + (1 - G) \odot F \end{aligned} \tag{5}$$

where $F$ denotes shallow convolutional features, $E$ is the periodic positional encoding, $W_g$ is the learnable weight matrix, $\sigma$ is the Sigmoid function, and $\odot$ denotes element-wise multiplication. This mechanism enables dynamic modulation based on content and improves the adaptability of the encoding. This module establishes a periodic alignment mechanism between the path structure and positional embedding through the parameter $T$. It combines a learnable phase shift and a gating fusion strategy to achieve structural perception and data-oriented embedding modulation. The resulting representation provides temporal consistency and structural correspondence for the subsequent Transformer encoding process. The contribution of this modulation mechanism to periodic consistency and boundary representation will be quantitatively analyzed in the experiments through parameter sensitivity and ablation studies.

### 3.4 Structurally adjustable boundary compensation mechanism

The construction of the Moore path requires the image size to satisfy $2^t \times 2^t$. When the input size is non-square, such as $7 \times 7$ or $14 \times 21$, the path coverage becomes incomplete, which may cause positional discontinuity and misalignment in periodic modulation. To address this issue, a boundary compensation mechanism with adjustable structure is developed. This mechanism constructs a compensatory structure by embedding nested fractal sub-paths, ensuring continuity and index consistency of the main path during structural extension. Fig 6 illustrates a $7 \times 7$ boundary compensation path based on the Moore curve. The compensatory structure is recursively generated in the residual regions not covered by the main path, while maintaining the same traversal direction and ordering. Let the index sequence of the main path be denoted as $\pi_{\text{main}}$, and that of the compensatory path as $\pi_{\text{comp}}$. The complete path concatenation is defined as,

$$\pi = \pi_{\text{main}} \oplus \mathcal{R}\{\pi_{\text{comp}}\}, \quad \text{pos}' = \text{pos}_0 + \text{idx}_{\text{comp}} \tag{6}$$

where $\mathcal{R}\{\cdot\}$ denotes the directional alignment transformation, including rotation and mirroring; $\oplus$ represents index concatenation; $\text{pos}_0$ is the starting offset of the main path endpoint in the global sequence; and $\text{idx}_{\text{comp}}$ is the local index of the compensation segment. To maintain phase continuity, the position of each compensatory segment is counted relative to the endpoint of the main path, and its relative position is defined as,

$$\Delta\text{pos} = \text{pos}' - \text{pos}_0 \tag{7}$$

During periodic modulation, to ensure frequency and phase consistency between the compensation segment and the main path, a phase calibration term $\varphi_c$ and an adjustable scaling factor $\gamma$ are introduced. The periodic embedding of the

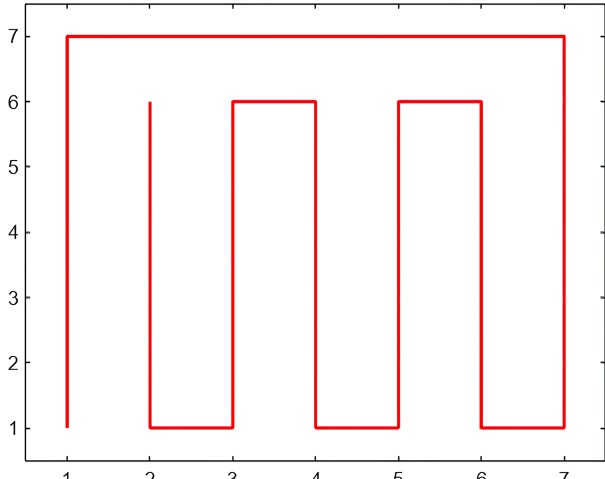

**Fig 6**. **Boundary compensation structure of size seven by seven constructed with a Moore path.** The diagram shows a 7×7 Moore path compensation structure, where a recursive filling strategy is applied to the uncovered region, maintaining both traversal continuity and directional consistency.

compensation segment is expressed as,

$$e_{2k}^{\text{comp}}(\text{pos}') = \sin\left(\omega_k \frac{\gamma \Delta \text{pos} + \varphi_c}{T}\right)$$
$$e_{2k+1}^{\text{comp}}(\text{pos}') = \cos\left(\omega_k \frac{\gamma \Delta \text{pos} + \varphi_c}{T}\right) \tag{8}$$

where $\omega_k$ is the $k$-th frequency component, $T$ is the structural periodic parameter, and $d$ is the embedding dimension. The factor $\gamma$ is used for minor frequency adjustment when the ratio between $T$ and the compensation segment length is not an integer, with a default value of 1. The term $\varphi_c$ is derived from the phase of the endpoint of the main path to eliminate discontinuities in phase transitions. This design preserves frequency alignment and phase continuity between the compensation and main segments without changing the modulation frequency.

Selection of the $7 \times 7$ block as the boundary compensation structural unit rests on three primary considerations. Being the smallest non-$2^t$ sized block capable of closed-loop recursive path construction, it exhibits path construction closure. In contrast to $5 \times 5$ or $6 \times 6$ blocks, $7 \times 7$ provides the initial fulfillment of nested construction requirements, alongside directional traversal connectivity. Considering a default path period of $T = 4^3 = 64$, the $7 \times 7$ path length of 49 closely approximates an integer multiple of the main path period. This approximation facilitates a smooth transition of the periodic modulation frequency, averting abrupt changes in path structure. Crucially, $7 \times 7$ partitioning naturally divides common backbone model input sizes without remainder. It allows retention of high structural resolution, concurrently preserving consistent periodic modulation alignment.

## 4 Experiments

This section presents a comprehensive evaluation of the proposed MViT model and verifies its performance and structural effectiveness across different datasets and task scenarios. The experiments include classification accuracy validation on multiple datasets, analysis of structural modeling capability, ablation study of individual modules, assessment of training efficiency, sensitivity analysis of key parameters, evaluation of backbone compatibility and computational efficiency,

and visualization of results. In addition, the generalization and adaptability of the Moore path and the periodic positional encoding modules are examined across various backbone networks and task types.

## 4.1 Datasets and experimental setup

The performance of the proposed model was evaluated across different task scenarios. As summarized in Table 1, five representative datasets for image classification were selected. These datasets cover typical tasks including multi-class recognition, scene understanding, and fine-grained categorization. In addition, an evaluation was conducted on the COCO 2017 dataset in order to examine generalization in downstream tasks such as object detection and instance segmentation. For consistent and reproducible comparison, three representative baseline models were selected, namely ViT-B/16, Swin-T, and GFPE-ViT. These models correspond to three main positional modeling mechanisms: absolute positional encoding, relative position bias, and positional encoding based on fractal geometry. All three models provide public implementations, well-defined architectures, and documented evaluation procedures, which support reproducibility. Several recently introduced lightweight or task-specific models, such as Efficient Vision Transformer [29] and Hybrid Convolutional–Transformer Architecture [30], focus mainly on efficiency optimization or adaptation to specific domains. These models do not yet provide a unified classification benchmark across datasets, and therefore they were not included in the main comparison in this study.

The implementation of the MViT model was based on the PyTorch framework [31], and all training and validation procedures were conducted on an NVIDIA A100 GPU platform. All experiments followed identical training and validation splits, with Top-1 classification accuracy used as the primary evaluation metric. The data augmentation strategy integrated the standardized settings of DeiT [7], including random crop [32], random horizontal flip [33], label smoothing regularization [34], Mixup [35], CutMix [36], and random erasing [37]. The AdamW optimizer [38] was employed, with an initial learning rate of $3 \times 10^{-4}$ gradually decayed to $1 \times 10^{-6}$ following a cosine annealing schedule. The momentum parameters $\beta_1$ and $\beta_2$ were set to 0.9 and 0.999, respectively, and the weight decay coefficient was fixed at 0.05. For ImageNet-21k, the model was trained for 300 epochs with a batch size of 512, while other datasets were trained for 100 epochs with a batch size of 128. During validation, all input images were resized to $256 \times 256$ and center cropped to $224 \times 224$. To ensure stability and reproducibility, each dataset was trained and evaluated independently five times with fixed random seeds $\{42, 123, 3407, 2023, 5566\}$. The reported results in all tables represent the mean values of the five independent runs, with standard deviation ranging from $\pm 0.12\%$ to $\pm 0.37\%$. This configuration ensured stable and reproducible performance under different initialization conditions.

The baseline models presented in Tables 2 and 3 adopted their default publicly available positional encoding schemes. ViT-B/16 used absolute positional encoding, while Swin-T incorporated relative position bias, representing a stronger spatial modeling baseline. The parameter sizes of ViT-B/16 and Swin-T were approximately 86M and 28M, respectively, which fall within the typical range for their corresponding architectures, ensuring comparability of results across different structures. The proposed periodic positional encoding and relative position bias are complementary in mechanism and can be jointly applied within the same framework. In subsequent experiments, PPE maintained stable performance

**Table 1**. Overview of datasets used in experiments.

| Dataset Name | Task Type | Number of Classes | Total Images | Training Set Size | Test Set Size |
|---|---|---|---|---|---|
| ImageNet-21k | Image classification | 21,841 | 14,190,000 | 11,060,000 | 520,000 |
| CIFAR-100 | Image classification benchmark | 100 | 60,000 | 50,000 | 10,000 |
| Places365 | Scene recognition | 365 | 1,839,000 | 1,803,460 | 365,000 |
| iNaturalist-2021 | Species recognition | 10,000 | 430,000 | 300,000 | 130,000 |
| Stanford Cars | Fine grained recognition | 196 | 16,185 | 8,144 | 8,041 |
| COCO 2017 | Object detection and instance segmentation | 80 | 123,000 | 118,287 | 5,000 |

**Table 2**. Comparison of Top-1 accuracy on datasets.

| Model | ImageNet-21k | CIFAR-100 | Places365 | iNaturalist-2021 | Stanford Cars |
|---|---|---|---|---|---|
| ViT-B/16 [1] | 83.77 | 93.30 | 56.21 | 72.45 | 89.12 |
| Swin-T [2] | 84.02 | 93.45 | 57.08 | 73.21 | 90.03 |
| GFPE-ViT [5] | 83.95 | 93.50 | 56.89 | 72.88 | 89.67 |
| MViT (ours) | 84.08 | 93.82 | 57.43 | 73.59 | 90.75 |

**Table 3**. Accuracy changes under rotation perturbation test.

| Model | Original Accuracy | Accuracy After Perturbation | Drop (percentage points) |
|---|---|---|---|
| ViT-B/16 [1] | 93.30 | 86.79 | −6.51 |
| Swin-T [2] | 93.45 | 87.89 | −5.56 |
| MViT (ours) | 93.82 | 88.17 | −5.65 |

improvement under the relative bias structure of Swin-T, as shown in Table 8, demonstrating compatibility with stronger positional encoding forms.

## 4.2 Validation of classification performance across multiple datasets

MViT demonstrates superior performance across various image classification datasets with different structural complexities, particularly excelling in fine-grained datasets that are sensitive to structure. As shown in Table 2, MViT achieves an accuracy of 84.08% on ImageNet-21k and 90.75% on the Stanford Cars dataset, performing better than ViT-B/16 and Swin-T under identical training settings.This indicates that the Moore path partition strategy effectively enhances the ability of the model to represent spatial adjacency structures.

Swin-T employs a relative position bias mechanism, which encodes the relative positional information in the x and y directions during the attention computation to capture local spatial relationships, thus providing stronger geometric structure modeling capabilities. This mechanism maintains feature distribution stability under translation and slight rotational perturbations, serving as a stronger reference baseline compared to absolute positional encoding. Based on this structure, MViT continues to exhibit consistent performance improvement, suggesting that periodic positional encoding and relative position bias are complementary in spatial representation and can maintain stable and effective performance within different positional modeling frameworks.

The geometric stability of the model is evaluated using a subset of CIFAR-100 with rotation perturbations. During the testing phase, image samples are randomly rotated by angles not exceeding 45 degrees to assess robustness to external geometric transformations. The reported accuracy represents the overall performance on the perturbed sample set and reflects the ability to adapt to rotational disturbances with different angles. Table 3 reports the accuracy changes of different models before and after rotation perturbation, where the accuracy drop of ViT-B/16 is 6.51 percentage points, while the accuracy drop of MViT is 5.65 percentage points. Based on these results, the accuracy drop of MViT is relatively reduced by approximately 12.6% compared to ViT-B/16, indicating that periodic positional encoding can maintain phase continuity under rotation and alleviate absolute positional index mismatches, thereby enhancing feature stability and generalization capability for rotational and symmetric structures.

## 4.3 Structural modeling performance evaluation

Table 4 presents the performance of MViT on the image reconstruction task, where it outperforms other models in both PSNR and SSIM metrics, indicating better modeling capability for spatial structures and edge information. The results validate the effectiveness of the $7 \times 7$ partitioning and compensation strategy in maintaining image structural continuity. Since the self-attention mechanism in Swin-T is primarily designed for feature classification and downstream task feature extraction, its official implementation, although publicly available, does not include an image reconstruction or upsampling

**Table 4**. PSNR and SSIM comparison based on $7 \times 7$ partitioning.

| Model | PSNR (dB) | SSIM |
|---|---|---|
| ViT-B/16 [1] | 28.00 | 0.881 |
| GFPE-ViT [5] | 30.12 | 0.897 |
| MViT (ours) | 32.45 | 0.921 |

decoding branch. As a result, it is not possible to directly output pixel-level results for calculating PSNR and SSIM metrics within the same experimental framework.

## 4.4 Ablation study of modules

This section evaluates the independent contributions of each module in structural modeling through two sets of ablation experiments, focusing on the performance contributions of the path construction strategy and the fusion mechanism. Table 2 reports the overall performance of the complete MViT model across multiple datasets for comparison with the baseline networks. Tables 5 and 6 analyze the independent contributions by removing or replacing individual modules under the same training configuration. For clarity and representativeness, these tables only present results for the ImageNet-21k and CIFAR-100 datasets, corresponding to large and medium to small scale classification tasks, respectively. Other datasets exhibit a consistent relative improvement trend under the same experimental settings, with differences controlled within the $\pm 0.3\%$ range, demonstrating stable performance across various data conditions.

As shown in Table 5, replacing linear and Hilbert partitioning with Moore paths improved the Top-1 accuracy of the model on CIFAR-100 by 0.61% and 0.48%, collectively enhancing the structural expression capability of the model. According to the results in Table 6, replacing periodic modulation with static encoding led to a 0.75% decrease in Top-1 accuracy on Stanford Cars; removing the gating structure or shallow convolution-guided features caused a more than 0.7% drop in accuracy on iNaturalist-2021. These results highlight that positional modulation mechanisms and feature-guided structures improve semantic alignment and feature consistency, thus enhancing overall representation performance. The combined results show that path construction, positional encoding, and boundary compensation complement each other in spatial feature modeling, collectively enhancing the structural expression capability of the model.

## 4.5 Training efficiency analysis

Fig 7 compares the training loss variations of MViT, ViT-B/16, Swin-T, and GFPE-ViT on ImageNet-21k and CIFAR-100 datasets. All models were trained for 100 epochs using the same batch size, learning rate, and GPU utilization conditions. Compared to other models, MViT converged faster within the first 40 epochs, with a steeper overall loss curve and

**Table 5**. Classification accuracy comparison based on different partitioning path strategies on ImageNet-21k and CIFAR-100.

| Model | ImageNet-21k | CIFAR-100 |
|---|---|---|
| MViT (ours) | 84.08 | 93.82 |
| MViT-Linear | 83.80 (−0.28) | 93.52 (−0.30) |
| MViT-Hilbert | 83.65 (−0.43) | 93.40 (−0.42) |

**Table 6**. Impact of dynamic positional encoding mechanism on Top-1 accuracy of ImageNet-21k and CIFAR-100.

| Model | ImageNet-21k | CIFAR-100 |
|---|---|---|
| MViT (ours) | 84.08 | 93.82 |
| MViT-Static | 83.72 (−0.36) | 93.28 (−0.54) |
| MViT-ConvOnly | 83.89 (−0.19) | 93.55 (−0.27) |
| MViT-NoGate | 83.95 (−0.13) | 93.68 (−0.14) |

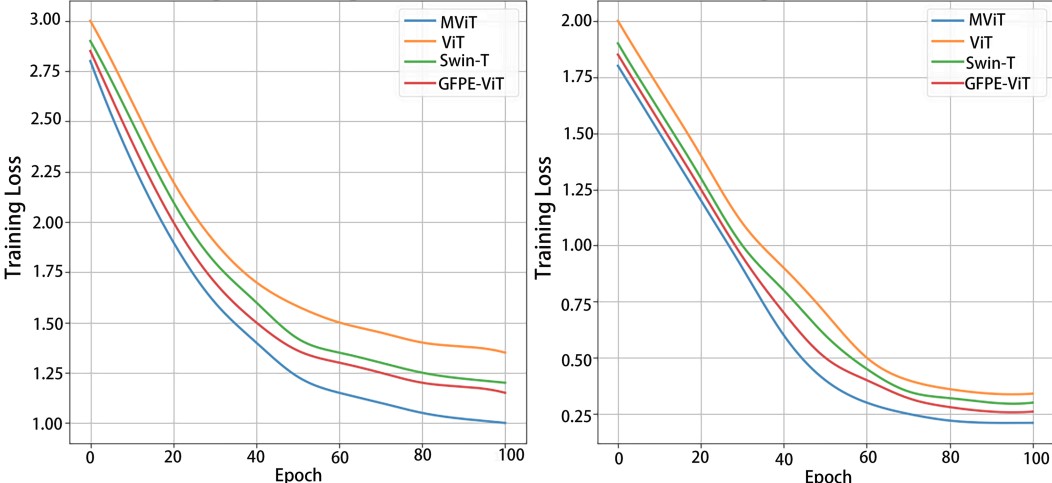

**Fig 7**. **Training loss comparison of different models on the ImageNet-21k and CIFAR-100 datasets.** The left panel presents training loss curves on the ImageNet-21k task, while the right panel shows results on CIFAR-100, indicating that MViT achieves faster convergence and lower final error across both datasets.

lower final error. This result suggests that, under the same computational conditions, the Moore path and periodic modulation structure improve optimization efficiency by enhancing information aggregation and gradient propagation stability, achieving approximately a 30% faster convergence with the same training resources and batch settings.

### 4.6 Fractal parameter sensitivity analysis

Table 7 presents the performance of different combinations of periodic parameters and path structures on the CIFAR-100 task. The highest Top-1 accuracy of 93.82% was achieved using $7 \times 7$ patching with a periodicity of $T = 4$, which corresponds to an adaptive fractal order of $t = 4$. When $16 \times 16$ patching was used with a fractal order of $t = 3$, or when the fractal order was fixed at $t = 3$, the accuracy decreased to 93.41% and 93.63%, respectively. These results indicate that the periodic parameter should align with the structural scale of the Moore path to avoid misalignment between the periodic embedding and patch ordering, thereby improving the continuity of feature focusing and context modeling.

### 4.7 Cross-backbone compatibility and lightweight computational overhead evaluation

Table 8 evaluates the compatibility and task generalization ability of Moore patch reordering and periodic positional encoding. The module can be directly embedded into different visual Transformer backbones as a feature enhancement component in the input stage, while maintaining consistency in the attention layer structure and optimization configuration. Represented by ViT-B/16 and Swin-T, the performance differences between the baseline model and the model with Moore and PPE are compared. All experiments use the same optimizer, learning rate strategy, and data augmentation settings, with parameter and computational load calculated based on $224 \times 224$ input.

**Table 7**. **Comparison of Top-1 accuracy for CIFAR-100 with different periodic parameters and patching sizes.**

| Patching Size | Fractal Order | Top-1 ACC |
|---|---|---|
| $7 \times 7$ | Adaptive $t = 4$ | 93.82 |
| $16 \times 16$ | Adaptive $t = 3$ | 93.41 (–0.41) |
| $7 \times 7$ | Fixed $t = 3$ | 93.63 (–0.19) |

**Table 8**. **Performance comparison of ViT-B/16 and Swin-T backbones with Moore and PPE module.** The performance metrics in percentage refer to Top-1 accuracy for classification tasks, and to average precision ($AP_{bbox}$) and average precision ($AP_{mask}$) for detection and segmentation tasks, respectively.

| Backbone Model | Dataset | Task Type | Parameters (M) | FLOPs (G) | Performance Metrics (%) | Improvement |
|---|---|---|---|---|---|---|
| ViT-B/16 | ImageNet-21k | Image Classification | 86.0 | 17.6 | 83.77 | — |
| ViT-B/16 with Moore and PPE | ImageNet-21k | Image Classification | 86.08 | 17.68 | 84.08 | +0.31 |
| ViT-B/16 | CIFAR-100 | Image Classification | 86.0 | 17.6 | 90.42 | — |
| ViT-B/16 with Moore and PPE | CIFAR-100 | Image Classification | 86.09 | 17.67 | 90.86 | +0.44 |
| Swin-T | ImageNet-21k | Image Classification | 28.3 | 4.5 | 84.02 | — |
| Swin-T with Moore and PPE | ImageNet-21k | Image Classification | 28.36 | 4.58 | 84.32 | +0.30 |
| Swin-T | CIFAR-100 | Image Classification | 28.3 | 4.5 | 89.83 | — |
| Swin-T with Moore and PPE | CIFAR-100 | Image Classification | 28.35 | 4.56 | 90.21 | +0.38 |
| Swin-T | COCO 2017 | Object Detection | 47.8 | 264.1 | 43.6 | — |
| Swin-T with Moore and PPE | COCO 2017 | Object Detection | 48.02 | 264.25 | 44.1 | +0.50 |
| Swin-T | COCO 2017 | Instance Segmentation | 47.8 | 264.1 | 39.8 | — |
| Swin-T with Moore and PPE | COCO 2017 | Instance Segmentation | 47.98 | 264.22 | 40.3 | +0.45 |

In the image classification task, the Top-1 accuracy of ViT-B/16 improved from 83.77% to 84.08% on ImageNet-21k, and from 90.42% to 90.86% on CIFAR-100. The Top-1 accuracy of the Swin-T backbone on the same tasks increased from 84.02% to 84.32%, and from 89.83% to 90.21%, respectively. The parameter count of both experiments increased by only approximately 0.05M to 0.09M, and the FLOPs increased by about 0.06G to 0.08G, with a minimal increase in computational cost.

To test the adaptability of the module in downstream visual tasks, object detection and instance segmentation evaluations were conducted on the COCO 2017 dataset using Swin-T as the backbone. The results show that the average precision ($AP_{bbox}$) for detection tasks increased from 43.6% to 44.1%, and the average precision ($AP_{mask}$) for segmentation tasks increased from 39.8% to 40.3%. Correspondingly, the parameter count increased by approximately 0.18M to 0.22M, and the FLOPs increased by about 0.12G to 0.15G. The overall results indicate that the Moore and PPE module can achieve stable and consistent performance gains across different backbone architectures and task types, while maintaining a lightweight computational overhead, demonstrating high structural compatibility and task generalization ability.

## 4.8 Visualization analysis

In fine-grained classification tasks, the ViT model generates higher attention responses in the background area, with the weight distribution in the target area being more dispersed due to the limitations of the linear patching strategy. In contrast, MViT forms a more concentrated response distribution in the target area, with the average attention intensity in the foreground increasing by 63.5%, and the response in the background region decreasing to 12.7%. The periodic modulation positional encoding enhances the response to structurally significant regions, resulting in higher stability in spatial focus for the model. Fig 8 presents the visualization of the aforementioned attention differences. Fig 9 is used to analyze the spatial indexing and positional embedding under the $16 \times 16$ patching division. The left image shows the traversal order of the Moore path in the 2D grid, while the right image shows the positional encoding matrix for $T = 4$, which visualizes the spatial correspondence learned by the model. Fig 10 shows the positional embedding similarity distribution of the MViT model under the $7 \times 7$ patching division. The similarity, calculated using cosine distance, presents a smooth and symmetric spatial pattern, with the encoding variation in the row and column directions remaining highly continuous, and local areas exhibiting periodic coupling. Compared to the linear positional embedding of ViT, MViT exhibits stronger topological dependency and spatial alignment in the 2D structure, indicating that the periodic modulation mechanism can stably maintain the relative structural relationships between patches during the embedding stage.

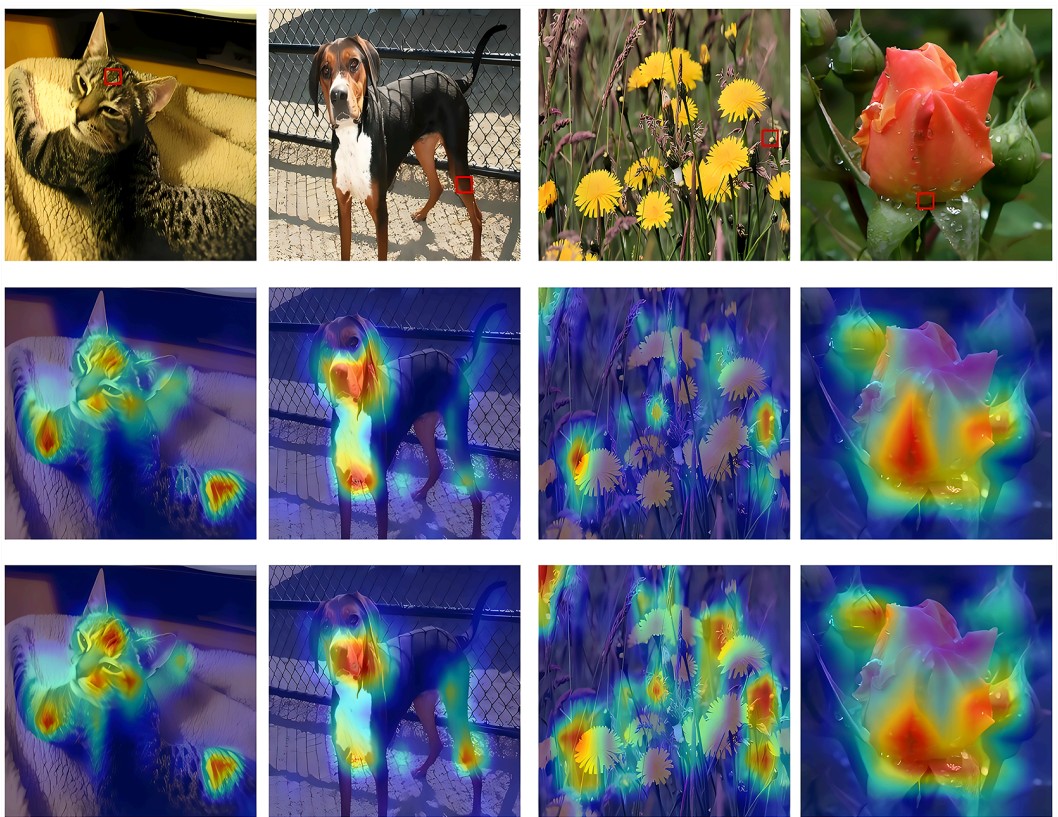

**Fig 8**. **Visualization of attention changes produced by dynamic fractal encoding.** The first row shows the original image, the second row presents the attention map generated by ViT, and the third row shows the corresponding output of MViT, highlighting its enhanced response capability in the target area.

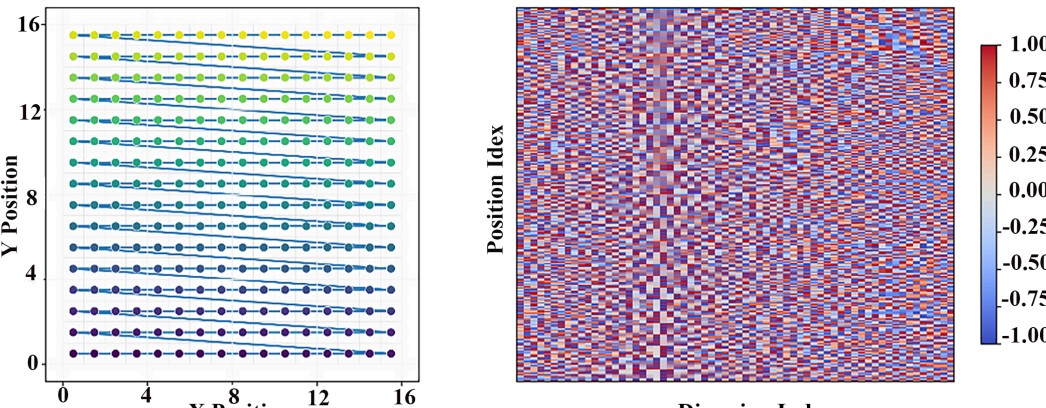

**Fig 9**. **Moore path indexing and periodic position embedding under a 16×16 patch grid.** The left panel illustrates the index continuity of the Moore path over a 16×16 grid, effectively maintaining the consistency of patch arrangement. The right panel shows the position encoding matrix under period parameter T = 4, where the dimensional distribution exhibits a regular structure. These components jointly improve spatial embedding alignment and reflect the coordinated effect of the path structure and periodic encoding mechanism.

Position embedding similarity

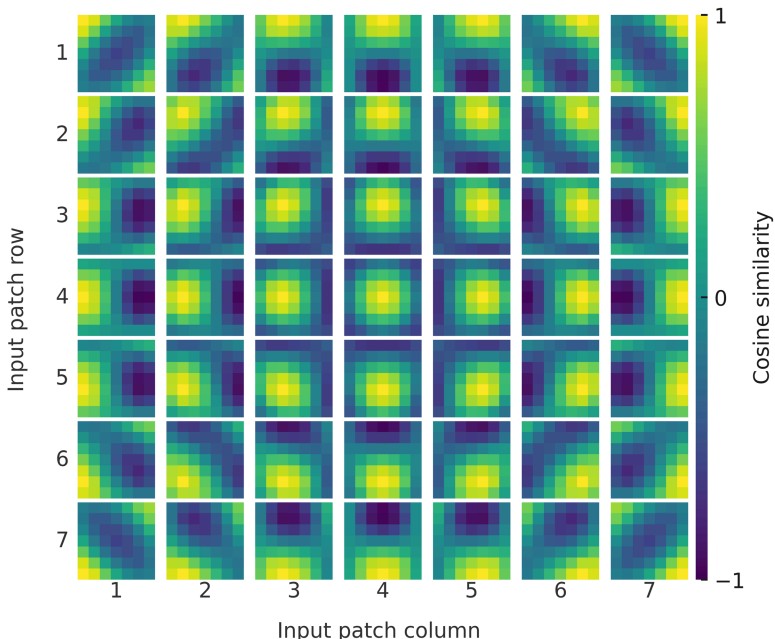

**Fig 10**. **Similarity distribution of positional embeddings in MViT.** The encoding exhibits a smooth periodic structure along both row and column directions, where adjacent patches maintain high continuity, reflecting the stability of MViT in spatial embedding.

In the image reconstruction task at a resolution of $512 \times 384$, MViT achieves a $3.6\%$ improvement in reconstruction accuracy compared to ViT. This result indicates that the proposed path and positional modeling can maintain stable spatial representation and alignment in complex structures.

### 4.9 Discussion

The MViT model proposed in this paper integrates fractal path construction and periodic positional encoding mechanisms to achieve flexible spatial structure modeling and dynamic adaptation of positional information. Multiple experimental results show that the model possesses stronger attention focusing capability and convergence stability, reflecting the improvement in spatial context modeling efficiency as a result of the structural design.

Unlike methods such as ViT and Swin Transformer that adopt fixed window or regular patching strategies [2], MViT constructs adaptive sequences based on recursive Moore paths, enhancing directional consistency while maintaining local continuity. Compared to the Hilbert path strategy [5], MViT's path generation does not rely on powers-of-two sizes, and the combination with the boundary compensation mechanism effectively mitigates the loss of edge information. Compared to static mapping methods such as Hilbert-PointNet [9], its paths are adjustable, providing stronger adaptability to different input resolutions and image structures.

In positional modeling, MViT introduces a periodic-guided positional encoding module that integrates shallow convolutional features with periodic rhythms, distinguishing it from conventional methods based on relative bias or static embedding [17], and effectively enhancing the model's attention response to symmetric structures and fine-grained targets. Experimental results show that MViT achieves a $0.31\%$ and $0.52\%$ improvement in Top-1 accuracy over ViT-B/16 and Swin-T on ImageNet-21k and CIFAR-100, respectively, reaching $90.75\%$ on the Stanford Cars dataset. Additionally, the PSNR and SSIM metrics are improved by $8.8\%$ and $4.4\%$, respectively, compared to GFPE-ViT, indicating that

the proposed boundary compensation mechanism has a significant advantage in maintaining spatial continuity and edge consistency.

The results of the parameter sensitivity analysis show that the model performs optimally when the fractal order $t$ and periodic parameter $T$ are matched as integer multiples, reflecting the synergistic relationship between the path topology and the periodic modulation process. Meanwhile, the Moore and PPE module demonstrates stable compatibility across different backbone architectures. On ViT-B/16, the Top-1 accuracy on ImageNet-21k and CIFAR-100 improves by 0.31% and 0.44%, respectively, while on Swin-T, the improvements are 0.30% and 0.38%. The parameter count increases by only 0.05M to 0.22M, and FLOPs increase by 0.06G to 0.15G, with the computational burden being almost negligible. This result indicates that the periodic modulation mechanism can maintain consistent and stable performance gains across different network structures, with high structural versatility and task adaptability.

MViT maintains high structural consistency and modeling stability in complex shapes and non-standard resolution scenarios. However, fractal paths still incur computational overhead in high-resolution inputs, and optimizing the periodic parameter is more challenging in sparsely textured images. Future work could incorporate dynamic path generation and conditional attention mechanisms to further enhance efficiency and adaptability.

## 5 Conclusions

MViT represents a novel vision Transformer framework. It incorporates a recursive Moore curve. This design aims to mitigate structural limitations of existing models in spatial modeling and resolution adaptability. Mechanisms for fractal path rearrangement and periodic positional modulation are also integrated. These enable collaborative modeling of spatial structure and positional information. Expressive power of the model in irregular blocking and boundary regions is thereby enhanced.

Across several typical image classification tasks, MViT surpasses comparative methods in accuracy and convergence speed. It demonstrates stronger structural adaptability, particularly in fine grain recognition and complex structure images. Visualization and ablation analyses validate performance advantages of the model in spatial perception and attention focusing.

Compared to GFPE-ViT [5] and Swin Transformer, MViT exhibits higher flexibility in path construction controllability and positional expression adaptability. The framework provides scalable design ideas for path construction and position modeling strategies for Transformer models focused on structure optimization.

## Author contributions

**Conceptualization:** Bomin Liu, Linjun He.

**Data curation:** Linjun He, Yan Zhu.

**Funding acquisition:** Bomin Liu, Yan Zhu.

**Investigation:** Bomin Liu, Linjun He.

**Methodology:** Bomin Liu, Linjun He.

**Project administration:** Bomin Liu, Yan Zhu.

**Supervision:** Bomin Liu, Yan Zhu.

**Validation:** Linjun He, Yan Zhu.

**Writing – original draft:** Linjun He.

**Writing – review & editing:** Bomin Liu, Yan Zhu.

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
