## [Decision Letter · Decision Letter 0]

17 Sep 2025

PONE-D-25-38700MViT Dynamic Positional Encoding Vision Transformer Guided by Recursive Moore CurvesPLOS ONE

Dear Dr. Zhu,

Thank you for submitting your manuscript to PLOS ONE. After careful consideration, we feel that it has merit but does not fully meet PLOS ONE’s publication criteria as it currently stands. Therefore, we invite you to submit a revised version of the manuscript that addresses the points raised during the review process.

We look forward to receiving your revised manuscript.

Kind regards,

Anil Yaman, Ph.D.

Academic Editor

PLOS ONE

Journal Requirements:

3. Please note that PLOS One has specific guidelines on code sharing for submissions in which author-generated code underpins the findings in the manuscript. In these cases, we expect all author-generated code to be made available without restrictions upon publication of the work. Please review our guidelines at https://journals.plos.org/plosone/s/materials-and-software-sharing#loc-sharing-code and ensure that your code is shared in a way that follows best practice and facilitates reproducibility and reuse.

“This work was funded by the Industry-university Cooperation and Collaborative Education Project of the Ministry of Education under Grant 231003221260825, and in part by the 2024 Graduate Education Reform Project of Shanghai Dianji University under Grant A102252401102029”

5. Thank you for uploading your study's underlying data set. Unfortunately, the repository you have noted in your Data Availability statement does not qualify as an acceptable data repository according to PLOS's standards.

Additional Editor Comments:

The reviewers agreed on the merits of the paper and proposed approach but suggested several improvements and clarifications. Please consider these comments and suggestions for improvements.

Please review and evaluate the publications recommended in reviewer comments to determine whether they are relevant and should be cited. There is no requirement to cite these works unless they are relevant.

Reviewer's Responses to Questions

**Comments to the Author**

1. Is the manuscript technically sound, and do the data support the conclusions?

Reviewer #1: No

Reviewer #2: Partly

Reviewer #3: Yes

2. Has the statistical analysis been performed appropriately and rigorously?

Reviewer #1: No

Reviewer #2: N/A

Reviewer #3: No

3. Have the authors made all data underlying the findings in their manuscript fully available?

Reviewer #1: Yes

Reviewer #2: Yes

Reviewer #3: Yes

4. Is the manuscript presented in an intelligible fashion and written in standard English?

Reviewer #1: Yes

Reviewer #2: Yes

Reviewer #3: Yes

5. Review Comments to the Author

Reviewer #1: In this manuscript, the authors proposed an MViT framework to enhance the spatial dependency modeling of traditional vision transformers. Moore curves are used to order the patches for better global continuity, and periodic positional encoding is designed accordingly to adapt to the Moore paths and partition. The proposed framework outperforms a few traditional ViT architectures on multiple image classification tasks, and the ablation study is performed to prove the effectiveness of periodic positional encoding. Despite the strengths mentioned above, I have the following major concerns.

1. The nature of the proposed method. The authors focused on improving patch ordering and encoding in the ViT architecture, without changing the downstream attention layers. Therefore, the proposed method can be used as an “add-on” module to a wide range of existing ViT architectures using traditional ordering and absolute encoding, rather than “replace” them. The authors should consider emphasizing that the proposed method can flexibly equip other ViTs with fancy attention block designs, which can bring additional predicting power. As a result, to present the experimental effectiveness, the authors should consider adding comparisons like “ViT vs. ViT+proposed Moore+PE”, “Swin vs. Swin+proposed Moore+PE”, etc. This could significantly extend the application.

2. Computer vision tasks. Related to (1), as general ViTs can be combined with the proposed method, the enhanced models can still be used as the backbone of different types of CV tasks. Nevertheless, the authors focused on image classification tasks in this manuscript. While Section 4.3 is related to image reconstruction, the data set and settings in Table 3 are not clear. The authors should consider adding data sets and experiments for more common CV tasks such as object detection and instance segmentation (e.g., on COCO data set) to show the general advantages of the proposed module.

3. Baseline models. In Table 2, all the 3 baseline models use traditional absolute positional encoding which is known to be weak in cases like image rotation. As the periodic encoding usage is a novelty in this paper, the author should consider adding a baseline model using stronger encoding methods as well (e.g., relative/rotary encoding [1]).

In addition, while the Swin-T selection for Swin transformer is clear, what is the specific size of vanilla ViT (e.g., L/16) used in Table 2 and 3? If the parameter numbers are not comparable between different models, the results won’t make sense.

[1] Heo, Byeongho, et al. "Rotary Position Embedding for Vision Transformer." European Conference on Computer Vision. 2024.

4. Error bar. In the result tables, most of the accuracy enhancement brought by the proposed method are within 1%. The authors should consider reporting the standard deviation of the performance metrics, to show the stability and reproducibility of the performance.

5. Writing. Please check the writing and fix the typos (e.g., excel -> excels in the abstract).

Reviewer #2: Title - MViT Dynamic Positional Encoding Vision Transformer Guided by Recursive Moore Curves

In this paper, the authors proposed is a recursive Moore curve based ViT framework (MViT).

Queries and Suggestions (Issues observed)

1. The title of this manuscript has not effectively captured the interest of a broader audience outside its niche area, which will limit its impact and reach, please revised

2. The introduction can be enriched by briefly describing the state-of-the-art in the title of this study and providing more recently related references to support the foundation of this study and provide a better context for the current research. I will recommend citing recently published related papers;

a. "Semantic Segmentation of the Lung to Examine the Effect of COVID-19 Using UNET Model," in Applied Machine Learningand Data Analytics. AMLDA 2022. Communications in Computer and Information Science, 2023.

b. "Vision transformers for automated detection of pig interactions in groups," Smart Agricultural Technology, vol. 10, p. 100774, 2025.

3. The problem that the author is trying to address in this paper is not clear yet, (clearer motivation for the study is required)

4. In 4-5- lines, authors should summarize the literature gaps identified before starting the methodology (Kindly relate your work in the context of existing work, what are the shortcomings of existing studies?)

5. Author should re-check the correctness of equations, especially from equations 3 to 8

6. More discussion of the results are required

7. Services of language experts are required (be mindful of tenses, typo errors)

8. Image quality needs to be improved (High resolution image suggested)

Reviewer #3: The authors propose 3 methods to increase image classification using Vision Transformers. Namely (1) image patch arrangement (2) period-aware positional encoding and (3) boundary compensation strategy.

They show on several benchmark image datasets that your top 1 accuracy has a small improvement compared to others.

The overall reading style of the paper is good and clear to me. Below I have a few comments / questions / suggestions to improve the paper / understand it better.

On page 2 you claim that you have 30% faster convergence speed (coming from figure 7 I believe).

- Does your method takes more time to train compared to the others?

- Does it takes more GPU memory then the others (model parameters)?

1. On page 2 you say: “ Visualization analysis showed the model’ s identification error for rotational targets decreased by 12.6%. To me this is not explained / supported in the paper. Could you follow up on this?

2. As you claim that your method is better on rotational , symmetrical objects, would it be an idea that you run an experiment for this? Could you find a dataset with mostly rotational , symmetrical objects, and compare your classification score compared to others?

3. The original VIT paper (https://arxiv.org/abs/2010.11929) has a figure (figure 7 in the middle) on their learned positional embeddings, by this they show what the embeddings have learned, and that they correspond to the image patch. Could be a nice additional to your paper as well.

4. You compare yourself to ViT[1] (2022), Swin-T[2](2021), GFPE-ViT[5] (2024). Aren’t there no other recent models as some of these are some years old.

5. As you have 3 methods, it would be nice to see the results of your method in isolation. I believe you try to show this in table 4 and 5, but the numbers are exactly the same as in table 2 where you have put all your methods. Can you clarify?

6. For all tables, it is unclear to me if your experiments have been run once or multiple runs (with seeds). I do not see any st. Dev scores as well. I cannot see if you are better compared to the others due to a lucky run now.

7. In table 3, Swin-T[2] model comparison is not taken into account. Why?

8. For table 4 and 5, you decide to only report on 2 datasets, not all. Why?

9. I think there is a type in table number 1, in the Places 365 row. The sum training set + test set is not equal to total images.

10. It would be interesting to see some more images being divided in different patches as examples.

6. PLOS authors have the option to publish the peer review history of their article (what does this mean?). If published, this will include your full peer review and any attached files.

Reviewer #1: No

Reviewer #2: No

Reviewer #3: No

---

## [Author Response · Author response to Decision Letter 1]

15 Nov 2025

To the Academic Editor

Dear Dr. Anil Yaman,

We sincerely appreciate the time and effort devoted by you and the three reviewers to the evaluation of our submission entitled “MViT: A Vision Transformer with Fractal Path Reordering and Dynamic Positional Encoding” (Manuscript ID: PONE-D-25-38700).Your constructive observations and professional guidance have greatly contributed to improvements in the conceptual framework, experimental design, and overall clarity of this study. Each comment has been examined with great care, and corresponding revisions have been completed in accordance with its academic significance.

A clarification is provided regarding a previous misunderstanding related to the supporting information figures (S1–S10) mentioned in the editorial notice. Initially, the guidelines for supporting information were misinterpreted, and the Supporting Information section preset in the LaTeX template provided on the official PLOS ONE website was mistakenly understood as a place where the captions of figures already cited in the main text should be relocated. It is now fully understood that the primary purpose of supporting information is to provide additional materials that are relevant to the study but not directly cited in the main text. For figures that are cited in the main text, their captions must be embedded within the manuscript and placed immediately after the paragraph where they are first mentioned, in complete accordance with the explicit requirements of the journal.

In the revised version, the logical structure and methodological details of the proposed model have been refined. Additional experiments and data analyses have been included to confirm the effectiveness and cross-architecture compatibility of the MViT framework. The Abstract, Introduction, Methods, and Results sections have been thoroughly revised to achieve clear consistency among research motivation, methodological innovation, and empirical evidence. The entire text has undergone professional English editing to ensure clarity, precision, and full conformity with PLOS ONE publication requirements. All equations, figures, and visual elements have been carefully reviewed and optimized for accuracy and readability.

We are deeply grateful for the rigorous editorial process and the valuable guidance provided by the PLOS ONE editorial board. These comments have not only enhanced the quality of the paper but also provided meaningful directions for future investigations. The current submission includes the revised version of the paper and a detailed point-by-point response document describing how every comment has been addressed.

In accordance with the data availability policy of the journal, the dataset and source code associated with this study have been deposited in the Figshare repository to facilitate public access and reproducibility. These materials are available at URL: https://figshare.com/authors/Linjun_He/22524461. In addition, a funding statement has been added to the revised manuscript, clearly declaring that the funders had no role in the study design, data collection and analysis, decision to publish, or preparation of the manuscript. This clarification ensures the independence and academic integrity of the research.

We again extend our sincere thanks for the careful consideration of this work. We hope that the revisions adequately resolve all concerns and that the paper now meets the academic standards expected by PLOS ONE.

With highest respect, we kindly invite you to review the revised version.

Yours sincerely,

Author Team

School of Design and Art, Shanghai Dianji University

November 15, 2025

---

## [Decision Letter · Decision Letter 1]

15 Dec 2025

PONE-D-25-38700R1MViT: A Vision Transformer with Fractal Path Reordering and Dynamic Positional EncodingPLOS One

Dear Dr. Zhu,

Thank you for submitting your manuscript to PLOS ONE. After careful consideration, we feel that it has merit but does not fully meet PLOS ONE’s publication criteria as it currently stands. Therefore, we invite you to submit a revised version of the manuscript that addresses the points raised during the review process.

We look forward to receiving your revised manuscript.

Kind regards,

Anil Yaman, Ph.D.

Academic Editor

PLOS One

Journal Requirements:

**Additional Editor Comments:**

Thank you for addressing the comments and questions of the reviewers. All reviewers found the responses and improvements satisfactory. One reviewer pointed out two questions. Could you please address them and asses if there is any revisions to be made in the text?

Reviewers' comments:

Reviewer's Responses to Questions

**Comments to the Author**

1. If the authors have adequately addressed your comments raised in a previous round of review and you feel that this manuscript is now acceptable for publication, you may indicate that here to bypass the “Comments to the Author” section, enter your conflict of interest statement in the “Confidential to Editor” section, and submit your "Accept" recommendation.

Reviewer #1: All comments have been addressed

Reviewer #2: All comments have been addressed

Reviewer #3: All comments have been addressed

2. Is the manuscript technically sound, and do the data support the conclusions?

Reviewer #1: Yes

Reviewer #2: Yes

Reviewer #3: Yes

3. Has the statistical analysis been performed appropriately and rigorously?

Reviewer #1: Yes

Reviewer #2: Yes

Reviewer #3: No

4. Have the authors made all data underlying the findings in their manuscript fully available?

Reviewer #1: Yes

Reviewer #2: Yes

Reviewer #3: Yes

5. Is the manuscript presented in an intelligible fashion and written in standard English?

Reviewer #1: Yes

Reviewer #2: Yes

Reviewer #3: Yes

6. Review Comments to the Author

Reviewer #1: The authors have addressed my concerns, and the experiments have been extended to more types of computer vision tasks. No more comments from my side.

Reviewer #2: The manuscript has has been improved and it is suitable for publication to the best of my knowledge

Reviewer #3: Dear authors,

Thank you for response to my questions, I find them satisfactory. I also liked that you added table 8 which I believe, shows the value of your approach. Nevertheless, I still have two questions below.

Question 1:

In your paper you mention"MViT demonstrates superior performance across various image classification datasets

with different structural complexities, particularly excelling in fine-grained datasets that

are sensitive to structure. As shown in Table 2, MViT achieved an accuracy of 84.08%

on ImageNet-21k and 90.75% on the Stanford Cars dataset, significantly outperforming

both ViT-B/16 and Swin-T."

But looking at table 2, I only see a small improvement. Could you clarify how significance was determined?

Question 2:

The sentence above table 3 states:

"Compared to ViT-B/16, MViT exhibited a 12.6%

reduction in accuracy drop, demonstrating that periodic positional encoding can

maintain phase continuity under rotation, alleviating absolute positional index

mismatches, and thus improving the stability and generalization ability of the model for

rotational and symmetric structures."

It was not clear to me how you found 12.6%.

7. PLOS authors have the option to publish the peer review history of their article (what does this mean?). If published, this will include your full peer review and any attached files.

Reviewer #1: No

Reviewer #2: **Yes:** Sunday Adeola Ajagbe

Reviewer #3: **Yes:** Bob Borsboom

---

## [Author Response · Author response to Decision Letter 2]

18 Dec 2025

PONE-D-25-38700

MViT: A Vision Transformer with Fractal Path Reordering and Dynamic Positional Encoding

PLOS ONE

To the Academic Editor

Dear Dr. Anil Yaman,

We sincerely thank you and the reviewers for the valuable time and careful evaluation devoted to our manuscript entitled "MViT: A Vision Transformer with Fractal Path Reordering and Dynamic Positional Encoding" (Manuscript ID: PONE-D-25-38700).

We have carefully read and fully understood the editorial decision and all comments from the reviewers, and have made corresponding revisions to the manuscript accordingly. With respect to the additional questions raised in the previous round of review, we have addressed them individually in the revised manuscript and provided detailed point-by-point responses in the accompanying "Response to Reviewers" document.

We sincerely appreciate the constructive comments and suggestions provided by the editor and reviewers. These comments have been very helpful in improving the clarity of presentation, logical organization, and overall academic rigor of the manuscript. We believe that the quality of the manuscript has been further improved through this revision.

We hereby submit the revised manuscript for your kind consideration and would be grateful if you and the reviewers could review it again. Should any further revisions be required, we would be pleased to make additional modifications.

Thank you again for your patience and support.

Yours sincerely,

Author Team

School of Design and Art, Shanghai Dianji University

December 18, 2025

Response to Reviewer 3

Dear Reviewer,

Thank you very much for your careful review of our manuscript in this round and for the valuable comments you have provided. We sincerely appreciate your recognition of our previous responses and have given careful consideration to the two additional issues you raised.

Regarding your question about the relationship between the magnitude of performance improvements in Table 2 and the use of the term "significant" , we have re examined the relevant text and revised the wording in the manuscript accordingly. In the revised version, expressions that could potentially lead to a misunderstanding of statistical significance have been explicitly avoided. The performance comparisons are now described in a more accurate manner under identical training settings, ensuring that the statements are consistent with the experimental results.

With respect to your concern about the unclear origin of the "12.6%" value reported in Table 3, we have added explicit clarification in the revised manuscript. This value is now clearly defined as the relative reduction in the accuracy drop of MViT compared to ViT B 16 under rotation perturbations. In addition, the specific calculation based on the experimental results has been provided in the text to avoid potential ambiguity. We have also clarified that this metric is intended to characterize the relative differences in performance degradation under geometric perturbations, rather than to indicate an absolute improvement in classification accuracy.

We sincerely thank you for your careful attention to the details of the manuscript and for your emphasis on precise and rigorous wording. Your comments have been highly valuable in improving the clarity, consistency, and overall academic quality of the paper.

Once again, we greatly appreciate the time and professional effort you have devoted to helping improve this work.

---

## [Editor Report · Decision Letter 2]

29 Dec 2025

MViT: A Vision Transformer with Fractal Path Reordering and Dynamic Positional Encoding

PONE-D-25-38700R2

Dear Dr. Zhu,

We’re pleased to inform you that your manuscript has been judged scientifically suitable for publication and will be formally accepted for publication once it meets all outstanding technical requirements.

Kind regards,

Anil Yaman, Ph.D.

Academic Editor

PLOS One

---

## [Editor Report · Acceptance letter]

PONE-D-25-38700R2

PLOS One

Dear Dr. Zhu,

I'm pleased to inform you that your manuscript has been deemed suitable for publication in PLOS One. Congratulations! Your manuscript is now being handed over to our production team.

Kind regards,

on behalf of

Dr. Anil Yaman

Academic Editor

PLOS One